# Effect of Magnesian-Expansive Components in Steel Slag on the Volume Stability of Cement-Based Materials

**DOI:** 10.3390/ma16134675

**Published:** 2023-06-28

**Authors:** Quanming Long, Qinglin Zhao, Wei Gong, Yuqiang Liu, Wangui Gan

**Affiliations:** 1State Key Laboratory of Silicate Materials for Architectures, Wuhan University of Technology, Wuhan 430070, China; 2School of Materials Science and Engineering, Wuhan University of Technology, Wuhan 430070, China; 3Baowu Environmental Technology Wuhan Metal Resources Co., Ltd., Wuhan 430081, China

**Keywords:** steel slag, magnesian-expansive component, granules, volume stability

## Abstract

Millimeter-scale magnesian refractory granules were found to be a unique magnesian-expansive component in steel slag. To systematically study the effects of these granular magnesian-expansive components on the volume stability of cement-based materials containing steel slag, an investigation of their existing forms and influence on the volume stability was conducted in this paper. The various-sizing waste–magnesium–chromate-based refractory brick (Mg-Cr brick) granules and different (FeO + MnO)/MgO ratios’ synthetic MgO·*x*FeO·*y*MnO ternary solid solutions granules were adopted to simulate magnesian-expansive granules by partially replacing manufactured sand in mortar. The 100 °C–3 h boiling and 213 °C–2 MPa–3 h autoclaving treatments were adopted as volume stability testing methods. The results indicated that whether Mg-Cr brick or MgO·*x*FeO·*y*MnO solid solution, the concentration of expansive stress and the anisotropy expansion came with the granular size rising weakening the volume stability of cement-based materials which contained magnesian-expansive granules, significantly. Meanwhile, this phenomenon resulted in the ineffectiveness of the single linear expansion rate when assessing the qualification of volume stability. Furthermore, it also changed the mortars’ failure mode from “muddy damage” to “break into blocks”. Especially, there is no volume stability issue when the MgO·*x*FeO·*y*MnO satisfied (FeO + MnO)/MgO ≥ 1.00. Considering the significant effect of the granular magnesian-expansive components on the volume stability of cement-based materials containing steel slag, it is imperative to enhance the detection of both MgO content and mineral existing forms in steel slag in practical applications. For recommendation, the threshold value of conducting autoclaved volume stability testing on steel slag should be set at MgO ≥ 3%. Furthermore, the qualification cannot be judged by the single linear expansion rate; the specimens’ appearance integrity and strength loss should also be noted.

## 1. Introduction

Steel slag is a type of industrial solid waste generated by steel-making plants during steelmaking, whose mineralogical composition is comparable to that of cement clinkers, mainly comprising calcium silicate, calcium ferrite, a multi-element solid solution of FeO, MgO, and other divalent metal oxides (RO-phase), free lime (*f*-CaO), and periclase (*f*-MgO) [1,2]. Although steel slag exhibits some hydration, its current usage in the cement and concrete industry is limited as a mineral admixture or alternative aggregate [3,4,5,6,7,8,9]. It is commonly believed that the delayed hydration of *f*-CaO in steel slag is the primary reason for its poor volume stability [7,10]. However, steel slag also contains magnesian-expansive components, namely, *f*-MgO and RO-phases [11,12,13]. The hydration of MgO leads to the formation of Mg(OH)_2_, which produces a 148% volume expansion [11,12,13,14,15]. These magnesian-expansive components significantly affect the volume stability of cement-based materials. The influence of steel slag on the stability of cement-based materials needs to be further studied when it is used as granular material.

At present, the control of magnesian-expansive components in steel slag is relatively loose. The control standard for magnesian oxide follows the Chinese standard set for MgO in Portland Cement (5%) [16]. In the practical application of steel-slag-based mineral admixtures, there is little inspection of magnesian-expansive components, for two reasons: first, the mineral admixtures are used in small quantities in cement-based materials; and second, after grinding, they undergo a “dilution effect”, resulting in uniform expansion. Within a certain range, the micro-expansion produced by expansive components reduces the shrinkage of the cement matrix.

However, with the utilization area expansion of steel slag, a particular phenomenon appeared, which is the replacement of natural aggregate by steel slag in cement-based materials. For these applications, if the steel slag granules have poor volume stability and expand at a fixed point, they may cause significant damage to the overall materials. Therefore, in addition to considering the effect of *f*-CaO on volume stability, the influence of magnesian expansion components needs to be studied in depth.

The magnesian-expansive components in steel slag originate from two main sources. One is the magnesian refractories corroded by molten steel [11], and the other is the addition of magnesian minerals as slagging agents [17]. During the high-temperature reconstitution process, the magnesian components will have different fates depending on the basicity of steel slag. Magnesian components in the low-basicity steel slag generally combine with silicate, while *f*-MgO is separated from silicate in the high-basicity steel slag due to an excess of CaO. This *f*-MgO partly forms some multi-element solid solutions with iron, manganese, and other divalent metal cations in steel slag, resulting in the formation of RO-phases [18], which account for approximately 30% [19,20] of the steel slag. Opinions regarding the effect of RO-phase on volume stability differ among scholars. Some believe that the RO-phase is inert and has no effect on volume stability, while others contend that the contribution of the RO-phase to volume stability is related to its actual composition [21]. Some researchers [22,23,24] investigated the expansion characteristics of RO-phase powder through artificial synthesis and discovered that the molar ratio of divalent metal ions to Mg mainly affects the volume stability. However, the safety molar ratio range of the granular RO-phase has not been studied, especially when natural aggregates are replaced by steel slag. It is not clear how this type of magnesian-expansive component affects the volume stability of cement-based materials, necessitating research on the expansive characteristics of magnesian-expansive components with the formation of granules.

Moreover, for China, the current volume stability testing methods in standards mostly focus on the expansion rate of specimens, which is also true worldwide (as shown in Table 1 [16,25,26,27,28,29,30,31,32,33,34,35]). However, for the granular-expansive component, the specimens’ expansion rate may not make sense for the uneven distribution and limited replacement of aggregates. In addition, the expansion of specimens caused by granular-expansive components could be anisotropy, which makes the linear expansion rate of the specimen pattern more chaotic. The water immersion test is more popular worldwide, but this mainly focuses on valuing the effects of *f*-CaO. Han [36], Chen [37], and Mao [38] also compared various standards and concluded that the water immersion test cannot thoroughly evaluate the expansive characteristic of steel slag, nor can the liner expansion rate.

With the expansive applications of steel slag, especially the replacement between natural aggregate and steel slag, the influence of the granular magnesian-expansive component on volume stability becomes more important. This paper aims to investigate the expansive characteristics of magnesian-expansive components in the form of granules in steel slag, starting from their source and existing forms in steel slag. The mortar bar was used as a specimen. Considering the limitations of the single linear expansion rate when used as a criterion of volume stability, the residual strength and strength change of specimens after volume stability testing will be paid attention to. Specifically, this study employs a waste-magnesium–chromate-based refractory brick (Mg-Cr brick) and synthetic RO-phase to partially replace manufactured sand for the preparation of cement mortar to simulate the impacts of magnesian-expansive granules in steel slag on cement-based materials’ volume stability.

## 2. Materials and Methods

### 2.1. Raw Materials

The raw materials used to prepare specimens in this research were mainly Ordinary Portland Cement (P·O 42.5) produced in Hubei, China, manufactured sand with a fineness modulus of 2.95, and a waste Mg-Cr refractory brick provided by a steel-making plant in Hubei, China. The physical performance and chemical composition of P·O 42.5 are presented in Table 2 and Table 3, respectively. The chemical composition of the waste Mg-Cr brick is shown in Table 3. Moreover, several chemicals with chemically pure grades were used to synthesize the RO phase: MgO, MnCO_3_, Fe_2_O_3_, and Fe powder (Sinopharm Chemical Reagent, Shanghai, China).

### 2.2. Sample Preparation and Testing

#### 2.2.1. Synthesis of MgO·*x*FeO·*y*MnO Ternary RO-Phase

MgO, Fe powder, Fe_2_O_3_, and MnCO_3_ were weighted in accordance with Table 4, and then mixed with absolute alcohol into paste. The mixing paste was stirred for at least 2 h to ensure it was fully and evenly mixed, then put into a vacuum drying oven at 40 °C for drying. The drying mixture needed milling into powder again and was filled into molds. Extra 10 kN pressure was applied and maintained for 2 min to prepare ϕ 50 mm × 10 mm cylinder specimens, which were sent for 3 h calcination in the GF14Q atmosphere furnace (Nanjing Boyuntong Instrument, Nanjing, China) with 100% N_2_ atmosphere protection to avoid oxidation. The synthetic RO phase would be crushed into granules smaller than 4.75 mm for the subsequent process. The preparation and experimental processes are shown in Figure 1.

#### 2.2.2. Preparation of Mortar Specimens

The waste Mg-Cr bricks were crushed, and the granules were classified into four grades in accordance with the granule size 2.36–0.6 mm, 0.6–0.3 mm, 0.3–0.075 mm, and below 0.075 mm (powder). Mg-Cr brick granules were adopted to partly replace manufactured sand with different replacement rates of 1%, 2%, 4%, and 6%, respectively. The water-to-cement ratio was 0.5 and the cement-to-sand ratio was 1:3. A 40 mm × 40 mm × 160 mm mortar bar was prepared. The RO-phase mortar was composed of cement paste and RO-phase granules where there was no manufactured sand. During preparation, the ratio of water to cement was 0.5, while the ratio of cement to RO-phase granules was 1.0, counted by weight. The specimen size was 20 mm × 20 mm × 20 mm. The fresh mortars were placed in a standard curing room for 1 d, then demolded and put into water with 20 °C curing for 28 d.

#### 2.2.3. Treating Condition

Two volume stability testing treatments were adopted in this study: boiled with 100 °C–3 h and autoclaved with 213 °C–2 MPa–3 h [23]. The boiled condition was provided by the FA-31A boiling box (Luda Test Instrument, Shanghai, China) and the autoclaved condition by YZF-2S autoclave (Tianjing Gangyuan Test Instrument, Tianjing, China).

#### 2.2.4. Linear Expansion Rate

The value of the linear expansion rate was calculated by Formula (1) [25].
β = (*l*_2_ − *l*_1_)/*l*_1_ × 100(1)

For the 20 mm × 20 mm × 20 mm specimens [18], the *l*_1_ was the mean distance between four parallel faces except for the molded surface and its opposite, which was measured after demolding. The *l*_2_ was the value measured after different curing ages or treatments. For the 40 mm × 40 mm × 160 mm specimens, the *l*_1_ was the initial length of specimens, and the *l*_2_ was measured by indicator-type comparator after 28 d water curing or other treatments.

#### 2.2.5. Microstructure Characterization

The broken specimen pieces were collected and soaked in absolute alcohol for 24 h after testing. Then, they were dried in a 40 °C vacuum drying oven for 24 h to prepare them for microstructure characterization.

Quanta FEG 450 SEM/EDS (Thermo Fisher Scientific, Waltham, MA, USA) with 20 kV acceleration voltage and a 5.0 spot size was used to observe the microstructure. Empyrean XRD (PANalytical, Almelo, The Netherlands) was used to determine the mineral phases, and the spectra were analyzed in the range of 2θ 10°–90°.

## 3. Results

### 3.1. The Existing Forms of Mg in Steel Slag

There are two primary sources of magnesian components in steel slag: one is derived from the magnesian refractory of the steel-making furnace, which is eroded by molten-steel during steelmaking, and the other is derived from the magnesian minerals in the slagging agents. After high-temperature reconstruction, Mg exists in steel slag in the following four typical forms (Figure 2).

The first form is undisturbed refractory granules with a large-scale and maintained shape, as shown in Figure 2a. The second is *f*-MgO separated from silicate in high-basicity steel slag due to an excess of CaO, as shown in Figure 2b, which is dispersed irregularly within the calcium ferrite phase or in plate-shaped C_3_S. The third is the RO phase, which is the solid solution of MgO, FeO, MnO, CaO, etc. [39], whose composition is variable. The typical composition can be characterized as MgO·*x*FeO, while the RO phases are classified into a Fe-rich phase (Figure 2d) and an Mg-rich phase (Figure 2e) based on whether x exceeds 1.00. The final form is silicate.

It is notable that the former three magnesian constituents in steel slag exhibit certain expansive characteristics. Of these, the influences of *f*-MgO powder and RO-phase powder on the volume stability during the utilization of steel slag have been studied by some researchers. However, the refractory granules appear to have a larger size (millimeter-scale) and better maintain their shape compared with the RO phase and *f*-MgO, as shown in Figure 2a. Relevant studies about whether the presence of MgO in this form in steel slag affects the volume stability of cement-based materials, and the corresponding influence rule, are still lacking. Therefore, the subsequent sections will focus on the effects of magnesian expansion granules introduced by the refractory and the cementitious materials’ volume stability.

### 3.2. Interaction between Molten Steel and Refractory Brick

Since the magnesian refractory brick of the steel-making furnace is the only source of refractory granules in the steel slag, it is essential to investigate the interaction between molten steel and magnesian refractory. Therefore, this section focuses on the analysis of waste Mg-Cr refractory bricks, which are replaced in steel-making furnaces. XRD, BSE, and SEM-EDS techniques are adopted; the mineralogical composition, microscopic morphology, and element distribution of waste Mg-Cr refractory brick are examined in detail at a microscopic level.

Based on the XRD analysis results shown in Figure 3b, it can be observed that the mineral phases of the original Mg-Cr refractory brick are magnesian oxide and magnesia-chrome spinel. Magnesian oxide is the primary crystalline phase. In accordance with the section image (Figure 3a), it is evident that the magnesian oxide plays the role of matrix, and that the well-crystallized magnesia-chrome spinel exists in the form of granules in Mg-Cr refractory brick. Additionally, it is observed that the Mg-Cr brick exhibits a distinct difference in appearance between the side in contact with the molten steel (corroded side) and the side not in contact with the molten steel (uncorroded side), as shown in Figure 3a. The side in contact with the molten steel appears in a black molten state, while the side far from the molten steel maintains its original morphology. To further investigate the chemical composition difference, SEM-EDS was employed to analyze its morphology and composition. The micro-morphology of the waste Mg-Cr brick is shown in Figure 4.

The micro-morphology and compositions of the two sides of the Mg-Cr refractory brick, the corroded side and the uncorroded side, exhibit distinct differences, as revealed by the secondary electron images and back-scattered electron images. The corroded side lacks crystallographic characteristics and presents a uniform melt morphology. The contrast is also uniform under BSE, indicating a relatively uniform element distribution. Conversely, the uncorroded side exhibits more obvious cubic granules, and the element distribution in BSE shows some enriched areas. Brighter contrasting regions indicate Fe element enrichment, according to the EDS results. The clean Mg-Cr brick is almost Fe-free. The Fe element gradually diffuses from the molten steel to the Mg-Cr brick upon contact with the molten steel, as shown in Figure 4f and Table 5. The field of vision presents a relatively obvious regional distribution, with a lighter contrast on the corroded side and darker contrast on the uncorroded side. The EDS results are shown in Table 5.

Based on the EDS results, it can be observed that the atomic ratio of iron and magnesian decreased from 4.63 to 0.40 as the distance increased from the corroded side to the uncorroded side. The Fe content in Mg-Cr brick was found to be higher in areas closer to the molten steel. There was a visible contrast between the high-Fe region and the clean Mg-Cr brick in the microscopic morphology. The high-Fe region appeared to be a single phase of MgO and FeO solid solution, similar to the RO phase. However, the FeO/MgO ratio of corroded Mg-Cr brick showed a maximum value of 4.63, quite different from the typical proportion of FeO/MgO in the RO phase in steel slag (nearly equal to 1.0). Moreover, only a thin layer (<10 mm) of the Fe-rich part of the waste Mg-Cr brick remained on the surface, while the rest of the part in contact with the molten steel was completely eroded and carried away. This phenomenon can be considered a “large-scale replacement” between the magnesian component in the Mg-Cr brick and the molten steel (Fe).

Furthermore, while observing the morphology of steel slag, large-scale magnesian refractory granules were discovered with a single granule size of nearly 5 mm, which confirmed the phenomenon of “large-scale replacement” that occurred in the process of molten steel eroding Mg-Cr brick. Despite being mixed with some small particles of C_2_S, the large-scale magnesian granules still maintained relatively clear boundaries. Occasionally, large-scale magnesian granules fall into the molten steel, and they still retain their original scale and composition during high-temperature reactions. This shows that the volume stability problem introduced by such large-scale magnesian granules is an essential concern in the recycling of steel slag. Its effects on mortar volume stability will be discussed in detail in the following sections.

### 3.3. Micro-Structure of Synthetic RO-Phase

The RO phase is commonly recognized as an infinite solid solution of MgO and FeO. However, during the steel-making process, other divalent metal oxides, such as MnO, also participate in the generation of the RO phase to form MgO·*x*FeO·*y*MnO ternary solid solution. The influence of the RO phase on the volume stability of cement-based materials is intimately linked to its reaction activity, which, in turn, is contingent on the doping of other bivalent metal oxides, excluding MgO, in the RO-phase. Hence, to delve deeper into the influence of the RO phase with varying compositions on the volume stability of cement-based materials, MgO·*x*FeO·*y*MnO RO phases with different molar ratios were synthesized using artificial means for subsequent testing.

Figure 5a shows the calcined cake after 1250 °C, which reveals that with the increase in the (FeO + MnO)/MgO ratio, the color of the calcined cake deepens, and the metallic texture strengthens. The XRD diffraction pattern of the synthesized sample (Figure 5b) indicates that with the increased doping of FeO and MnO in the solid solution, the diffraction peak of the RO phase shifts towards a smaller angle direction, which can be attributed to the larger ionic radius of Fe^2+^ and Mn^2+^ compared with Mg^2+^. As the doping increases, the lattice constant expands, causing the diffraction angle to shift in a smaller angle direction. This regular shift in the diffraction peak can be utilized as an indicator for a successful RO-phase synthesis.

### 3.4. Effect of Refractory Granules on Mortar’s Volume Stability

As previously mentioned, some of the magnesian refractory granules remained in the steel slag after undergoing the “large-scale replacement”. These granules retain their original shapes and have a larger size. With the increased attention being paid to the comprehensive utilization of steel slag resources, steel slag is no longer solely used as an admixture but converted into steel slag sand and steel slag aggregate to replace natural aggregate. When using steel slag aggregates, the volume stability is a significant concern that must be addressed. However, the influence of the magnesian component granule size on the volume stability of steel slag remains unexplored. Therefore, waste Mg-Cr bricks from a steel-making plant were employed to create Mg-Cr brick granules with different granule sizes, and to replace a portion of the manufactured sand. The effect of different granule sizes and replacement amounts of Mg-Cr brick granules on the volume stability of cement mortar are discussed. Four Mg-Cr brick granule replacement levels of 1%, 2%, 4%, and 6% were adopted. The influence of Mg-Cr brick granules on mortar stability was examined across four granule sizes, ≤0.075 mm, 0.3–0.6 mm, and 0.6–2.36 mm.

#### 3.4.1. Effect of Replacement

The strength changes and linear expansion rates of mortar specimens with different waste Mg-Cr brick granules’ replacement after boiled and autoclaving treatment are presented in Figure 6 and Table 6, respectively.

Observations from Figure 6 indicate that the strength changes in mortar, which contain varying-size waste Mg-Cr brick granules, follow distinct patterns as the replacement increases. After boiling at 100 °C for 3 h, the mortar that contained ≤0.075 mm Mg-Cr brick granules exhibited a decrease in losses of strength with the increase in the replacement. The 6% replacement results in a lower strength loss than the blank group after boiling. This suggests that the expansion characteristics of the ≤0.075 mm Mg-Cr brick granules after the reaction with water not only do not damage the mortar, but also play a role in strengthening.

In contrast, for mortar samples containing Mg-Cr brick granules of granule sizes 0.3–0.6 mm and 0.6–2.36 mm, the strength loss consistently increases with the increase in replacement. Furthermore, when the replacement exceeds 1%, the mortar’s strength loss always exceeds that of the blank group, while the testing specimens are intact in appearance. In the case of the autoclaving treatment, the mortar’s strength loss is much more severe. Apart from the scenario where the replacement of Mg-Cr brick granules with ≤0.075 mm granules is 1%, the other testing specimens suffer damage and completely lose their structural strength after the autoclaving treatment. Additionally, the strength loss in the mortar, which contained 1%, ≤0.075 mm Mg-Cr brick granules, is slightly greater than that of the blank sample.

Compared with the strength loss, the mortars’ linear expansion rates of different Mg-Cr brick granules’ replacements are chaotic after the boiling treatment. Additionally, the change in the linear expansion rate of the testing specimens with varying amounts of substituted Mg-Cr brick granules is highly unpredictable. The expansion of waste Mg-Cr brick grains is anisotropic, resulting in a more irregular impact on the expansion rate of mortar specimens after the boiling treatment. This effect is even more pronounced under autoclaved conditions. Only mortar specimens that contained 1%, ≤0.075 mm Mg-Cr brick granules maintain their integrity in appearance.

#### 3.4.2. Effect of Granule Size

In this section, a comparison is made regarding the influence of Mg-Cr brick granules of varying granule sizes on mortar’s volume stability, maintaining the same replacement of 6%. The corresponding strength, strength loss, and linear expansion rates are shown in Figure 7.

From the perspective of strength, it was observed that, following 28 days of water curing, mortar specimens containing Mg-Cr brick granules displayed varying degrees of increased strength in comparison to the blank group. Notably, the group with ≤0.075 mm Mg-Cr brick granules exhibited the greatest increase in strength, indicating that some of the Mg-Cr brick granules reacted during the water curing. However, after the boiling treatment, it was observed that each group experienced a strength loss, and with the increase of the particle size of Mg-Cr bricks, the degree of strength loss increased. This suggests that the greater the granule size, the greater the damage caused by the magnesian-expansive component. Notably, regardless of the granule size, the testing specimens with 6% replacement are unable to maintain their appearance integrity after the autoclaving treatment. Furthermore, it can be found that the damage forms differ with variations in Mg-Cr brick sizes. Waste Mg-Cr brick grains of ≤0.075 mm cause muddy damage to mortar, while larger-scale waste Mg-Cr brick grains cause mortar to break into blocks.

The finer waste Mg-Cr brick granules achieved better dispersion within the mortar during the mixing process. Consequently, when damage occurs, it results in multi-point damage to the cementitious paste, ultimately converting the matrix paste to a muddy paste. At the same dosage, the larger waste Mg-Cr brick granules are distributed at some structural points in the mortar due to their larger size. Therefore, when their expansion characteristics are determined, the destruction is caused by “point damage”, which breaks the specimen into blocks.

### 3.5. Effect of RO-Phase Granules on Mortar’s Volume Stability

As noted previously, a part of the magnesian-expansive components in steel slag takes the form of an RO phase. The influence of the RO phase on the volume stability of the cement-based materials was not fully understood when steel slag was used as an aggregate. To address this knowledge gap, the influences on the volume stability of cement-based materials caused by MgO·*x*FeO·*y*MnO (RO phase) granules, which were artificially synthesized in the laboratory and used as an aggregate, are discussed in the following section. Figure 8 shows the strength, strength loss, and linear expansion rate of mortar, which contained RO-phase granules with different compositions, after different treatments.

As shown in Figure 8a, the strength of mortar containing varying RO phases is basically uniform after 28 d water curing, even compared with the blank group. However, the subsequent boiling treatment caused a decline in strength. This drop in strength decreased with the increasing (FeO + MnO)/MgO ratio, due to the change in the crystal structure of the RO phase from active periclase to inert wustite with the increase in the doping of Fe^2+^ and Mn^2+^ in the RO- phase. Consequently, the possibility in the RO phase of low activity exhibiting expansibility under the accelerated hydration treatment is lowered.

Figure 8b illustrates that the strength loss of the testing specimens after boiling was approximately 30%, higher than that of the blank group. This suggests that the boiling treatment causes some damage to the mortar. Nevertheless, the strength loss is slightly higher than that of the blank group, which means that it is within an acceptable range.

Figure 8c presents the linear expansion rates of the testing specimens after the boiling treatment and the autoclaving treatment. The figures show that the linear expansion rate of the testing specimens decreases with the increase in (FeO + MnO)/MgO, which is consistent with the pattern observed for the strength loss. However, it is worth noting that the linear expansion rate of the blank group after the autoclaving treatment is even lower than that after the boiling treatment. Additionally, the linear expansion rates of the testing specimens containing RO-phase granules after two treatments do not exhibit a significant difference, which makes it difficult to determine whether the testing specimens are damaged in accordance with the single linear expansion rate. Section 3.4 notes that the changes in the linear expansion rate of mortars that contained magnesian-expansive granules are irregular after the boiling treatment or the autoclaving treatment. Therefore, compared with the strength loss, the single linear expansion rates of the testing specimens cannot serve as a reliable criterion to determine the volume stability of cement-based testing specimens that contained magnesian expansion granules.

Similarly, during the appearance inspection of the testing specimens, it was observed that the failure modes of the testing specimens containing RO-phase granules after the autoclaving treatment also broke into blocks. As shown in Figure 8d, after the autoclaving treatment, the testing specimen with (FeO + MnO)/MgO = 0.34 was completely damaged, while the testing specimen with (FeO + MnO)/MgO = 0.60 had broken corners. Other testing specimens maintained an intact appearance, but the remaining strength was little. Only the residual strength of the testing specimen, which satisfied (FeO + MnO)/MgO = 1.00 in the RO-phase, was close to that of the blank group after the autoclaving treatment. Additionally, after the boiling treatment, the testing specimens with a low (FeO + MnO)/MgO ratio exhibited broken corners and other defects, which is similar to the damage that Mg-Cr brick granules caused to mortar, as described in the previous section. Moreover, the strength loss decreases with the increase in (FeO + MnO)/MgO ratio. The strength loss of the testing specimen is basically the same as that of the blank group when the (FeO + MnO)/MgO = 1.00. That is to say, the RO phase that satisfied (FeO + MnO)/MgO ≥ 1.00, even in granule form, will not affect the volume stability of the mortar.

The expansive granules at the structural points are responsible for these phenomena. Due to their large size, the expansion components in granule form are more likely to cause uneven “point damage” to the testing specimens after their expansion characteristics are displayed, especially for the highly rigid cement-based materials with a low acceptance rate for deformation. Furthermore, the granules located at the corners are also likely to cause significant damage to the appearance of the testing specimens, such as broken corners.

In the academic community, the activity of the RO phase is generally distinguished by the doping ratio of other bivalent metal oxides to MgO; that is, if the total doping ratio of other metal oxides is greater than 1.00, the RO phase is considered an inert wustite phase, while if the doping ratio is less than 1.00, the RO phase is considered an active periclase phase. The expansion characteristics of the active RO phase can cause cement-based materials to have poor volume stability. Based on the tests mentioned above, when (FeO + MnO)/MgO ≥ 1.00, the cement-based material containing an RO phase can still pass the autoclaved volume stability test, even when the RO phase is presented in granule form. It is important to clarify that the RO phase in this study was synthesized using a high-temperature sintering process. By using the dense RO phase with high mechanical strength as granules, the cement matrix’s strength loss after the autoclaving treatment is partially compensated, so the testing specimens have slightly higher residual strength compared with the blank group.

## 4. Discussion

In the industry, the influence of magnesian expansion components in steel slag on the volume stability of cement-based materials is relatively neglected compared with *f*-CaO. In accordance with the statements of GB/T 20491-2017 [16] and GB/T 51003-2014 [31], autoclaved volume stability testing can only be conducted when the MgO content in steel slag is greater than 5% and 13%. However, the testing results in this study indicate that the MgO introduced by 1% Mg-Cr brick granules is undoubtedly less than 5%, and the linear expansion rate of the corresponding testing specimen is also far less than the value specified in the standard of 0.5% after autoclaving treatment. However, the strength loss of the mortar is higher than that of the blank group, indicating poor volume stability. Additionally, after converting 1% manufactured sand into the replacement for powder, the value is 2.88% (rounded to 3%).

This indicates that the volume stability problem caused by magnesian-expansive components in steel slag cannot be ignored. It is also necessary to make appropriate adjustments to the current judging standards regarding whether to carry out an autoclaved volume stability test. Based on the preliminary exploration in this study, the detection limit should be reduced to at least MgO ≥ 3%, and further experimental research is required to obtain a more accurate detection limit.

For Mg-Cr brick granules, the research in this study found that the allowable limit content of Mg-Cr brick granules in the aggregate is less than 1%. Therefore, when steel slag is used as an aggregate, regardless of the MgO content in steel slag, an autoclaved volume stability test should be conducted as long as it contains MgO; otherwise, potential stability hazards cannot be eliminated.

The reactivity of the RO phase is another area of concern. Previous studies have indicated that when (FeO + MnO)/MgO ≥ 1.00 is in the RO phase, the RO phase is inert and does not pose a potential stability hazard. However, those studies have primarily focused on powder form; the influence of larger-scale RO-phase granules on cement-based materials is not well studied. It is uncertain whether the further “delayed hydration” resulting from the smaller specific surface areas of granules will have additional effects on the specimens. Nevertheless, the findings of this study confirm that the threshold value of (FeO + MnO)/MgO ≥ 1.00 remains applicable, even if the RO phase is presented in granule form.

Furthermore, this study also reveals that different granule sizes of magnesian-expansive components result in different mortar failure modes. Powder-form (≤0.075 mm) expansive components primarily destroy cementitious paste, leading to muddy damage to specimens. In contrast, granular (≥0.075 mm) expansive components cause the testing specimens to break into blocks, as shown in Figure 9. The dispersion effect of powder on cement-based materials is superior, resulting in the appearance of more destruction points and muddy damage characteristics, whereas the granules in cement-based materials cause destruction at a few points, resulting in the specimens breaking into blocks. The powder in the testing specimen is manifested as homogeneous expansion. The granular component exhibits anisotropic expansion, causing a relatively concentrated expansion at a specific point. Consequently, this type of expansion is more likely to cause damage to the appearance of the testing specimen, such as broken corners. Before it is damaged, the specimen will not show obvious expansion, which is also consistent with the fact that the volume stability of the mortar specimens containing waste Mg-Cr brick granules cannot be completely characterized by the single linear expansion rate.

In comparison with powder, granular expansion components cause greater harm to cement-based materials. When the tensile strength of the matrix is sufficient to bind the expansive stress, the testing specimen does not show strength loss, and micro-expansion can improve the strength. However, if the tensile strength of the matrix is less than the expansive stress, the testing specimen is damaged; its failure mode is shown in Figure 9, which indicates the muddy damage to the paste. In contrast, the granular-expansive components occupy more space, whose expansive stress is not uniformly distributed in the sample matrix, along with expansive components such as powder, while the stress concentration occurred at the specific points occupied by granules. This is also the reason why larger Mg-Cr brick granules increase the strength loss of the specimens and trigger them to break up easily. And the linear expansion rate is irregular under the same replacement amount, similar to the specimens containing RO-phase granules. Therefore, when using the mortar bar method to detect the volume stability of steel slag, the single linear expansion rate should not be the sole criterion. The appearance integrity and strength loss of the testing specimen are more effective criteria.

## 5. Recommendation

Our study found that the magnesian-expansive granules in steel slag significantly weakened the volume stability of cement-based materials; thus, the volume stability issue must be addressed in construction applications. For recommendation, the usage of magnesian refractory should be reduced in the steel-making furnaces to cut off the origin of magnesian-expansive granules. Feasible approaches include reducing the magnesian parts’ content in refractories or adopting new refractory types like aluminous and carbon-based. Furthermore, the volume stability test of steel slag must be conducted, especially when it is used to replace natural aggregates. As we have mentioned, the threshold value determining whether autoclaved volume stability testing is conducted should be reduced to MgO ≥ 3% in steel slag, at least. However, this value (MgO ≥ 3%) is just suppositional data; the specific value must be determined after a comprehensive comparison. For the volume stability test methods, the linear expansion rate cannot fully evaluate the stability of specimens containing granular-expansive components; thus, more evaluation criteria should be adopted, such as the appearance integrity of testing specimens and strength loss after testing.

## 6. Conclusions

The existing form of magnesian-expansive components in steel slag and the influence on cement-based materials’ volume stability according to their expansive characteristics are investigated in this study. Based on the analysis, the following conclusions can be drawn:(1)Except for *f*-MgO and RO-phases, the magnesian-expansive components in steel slag also include millimeter-scale magnesian refractory granules due to the “large-scale replacement” between the molten steel and the magnesian refractory of the furnace, which significantly affects the volume stability of cement-based materials containing steel slag.(2)After partially replacing manufactured sand with different-sized Mg-Cr brick granules and adding them to mortar, their expansive characteristics have a significant influence on the strength and volume stability of the specimens. At 28 d, the mortar’s strength loss increases with the increase in waste Mg-Cr brick granules’ replacement after 100 °C–3 h boiling treatment; after 213 °C–2 MPa–3 h autoclaving treatment, only the specimen containing 1%, ≤0.075 mm Mg-Cr brick granules can maintain its appearance integrity. Additionally, the strength loss rate of the specimens after the boiling treatment increases with the increase in granule size, which implies that large magnesian-expansive granules can significantly affect the volume stability of cement-based materials and even a small mass fraction of such components can completely destroy the specimens, especially compared with powder magnesian-expansive components.(3)The damage caused by magnesian-expansive granules to the testing specimens is different from that caused by powder. The expansion of powder is uniform, leading to muddy damage to cementitious paste. However, the granules cause expansive stress concentration, resulting in specimens breaking into blocks, while the linear expansion rate is irregular. The damage caused by magnesian expansion granules can be better characterized by the strength loss and appearance integrity of the testing specimen after the accelerated treatment.(4)The expansive properties of RO-phase granules are associated with the molar ratio, presented as (FeO + MnO)/MgO in this study. With the increase in (FeO + MnO)/MgO, the harm inflicted on the testing specimens by the hydration expansion of the RO phase gradually decreases. When (FeO + MnO)/MgO ≥ 1.00, the testing specimen containing RO-phase granules can be qualitatively detected by the autoclaved volume stability testing. Likewise, due to the expansive stress concentration caused by the granules, the linear expansion rate of the testing specimen is also irregular. The damage caused by the expansion of RO-phase granules can be more accurately indicated by the strength loss of the specimen after the accelerated treatment.(5)Currently, the testing standard for the volume stability of magnesian-expansive components in steel slag requires improvements, particularly regarding the high threshold of MgO content in the detection process. The threshold value determining whether autoclaved volume stability testing is conducted should be adjusted to MgO ≥ 3% in steel slag for reference in China.

## Figures and Tables

**Figure 1 materials-16-04675-f001:**
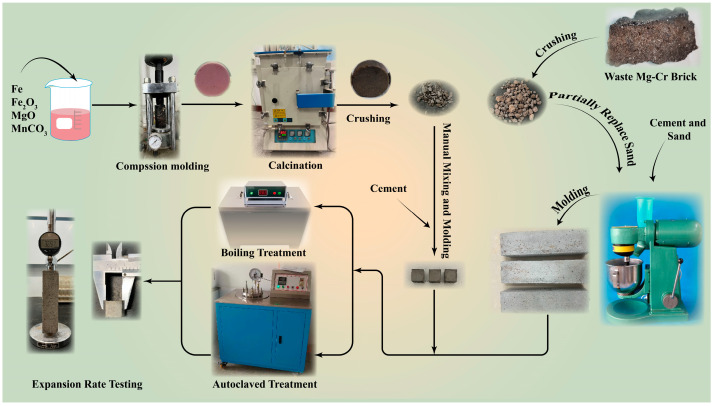
Preparation and experimental processes.

**Figure 2 materials-16-04675-f002:**
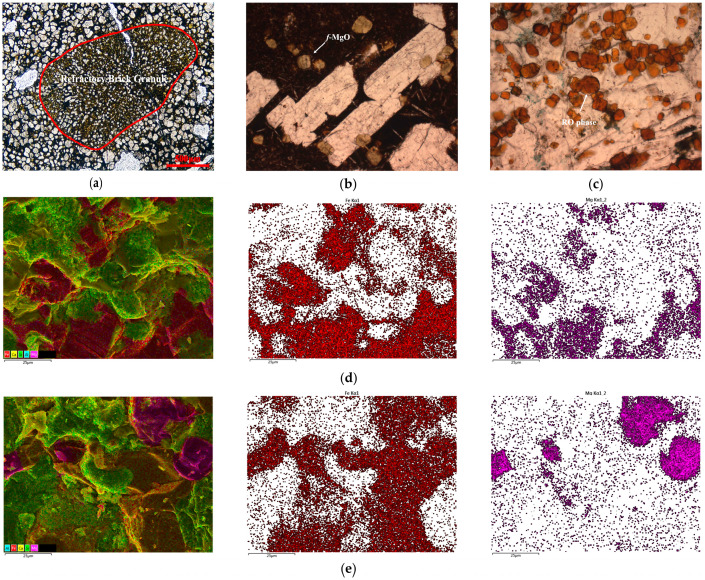
The existing forms of magnesian components in steel slag. (**a**) Refractory brick granule with maintained shape, polarization. (**b**) *f*-MgO, polarization. (**c**) RO-phase, polarization. (**d**) Fe-rich RO-phase (**left**) and the element distribution of Fe (**middle**) and Mg (**right**). (**e**) Mg-rich RO-phase (**left**) and the element distribution of Fe (**middle**) and Mg (**right**).

**Figure 3 materials-16-04675-f003:**
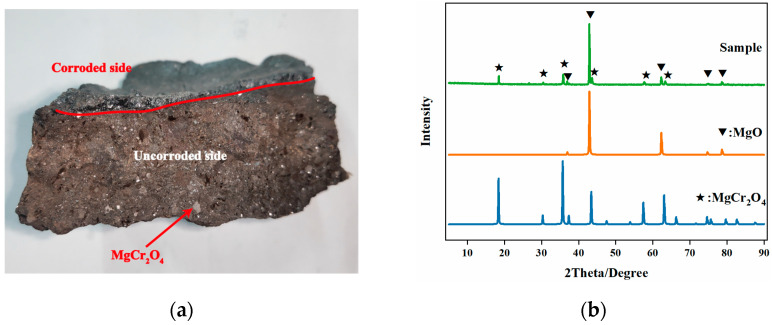
Longitudinal section (**a**) and XRD pattern (**b**) of waste Mg-Cr brick.

**Figure 4 materials-16-04675-f004:**
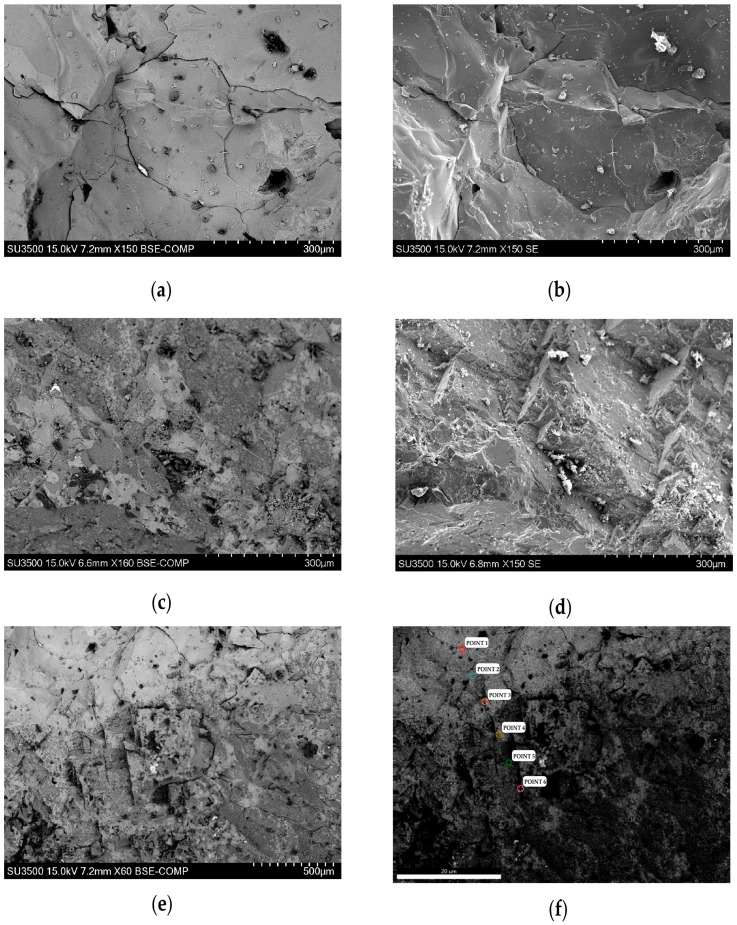
SE-BSE, SEM-EDS of waste Mg-Cr refractory brick. (**a**) BSE of corroded side. (**b**) SE of corroded side. (**c**) BSE of uncorroded side. (**d**) SE of uncorroded side. (**e**) BSE of waste Mg-Cr refractory brick. (**f**) EDS position.

**Figure 5 materials-16-04675-f005:**
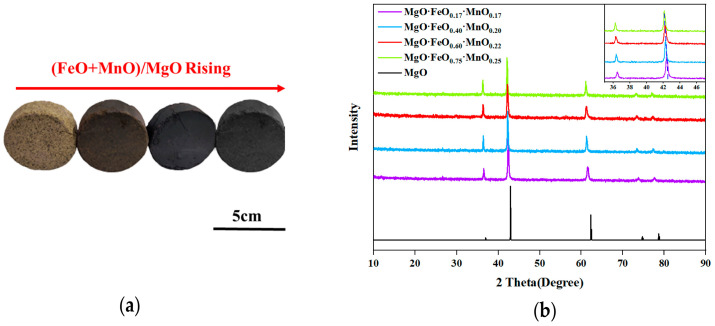
Synthetic RO phases (**a**) and their XRD pattern (**b**).

**Figure 6 materials-16-04675-f006:**
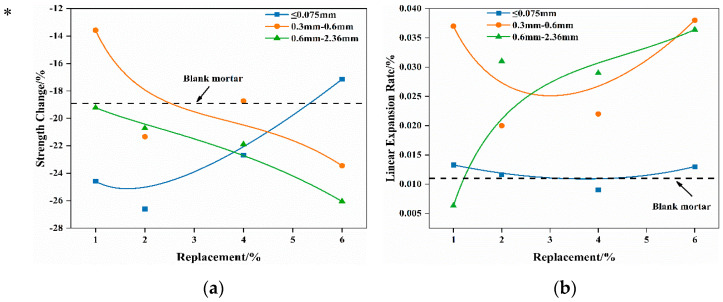
Strength change and linear expansion rate of mortars with the addition of waste Mg-Cr brick granules after the boiling treatment. (**a**) Boiled strength change. (**b**) Boiled linear expansion rate. * Testing specimens are intact in appearance.

**Figure 7 materials-16-04675-f007:**
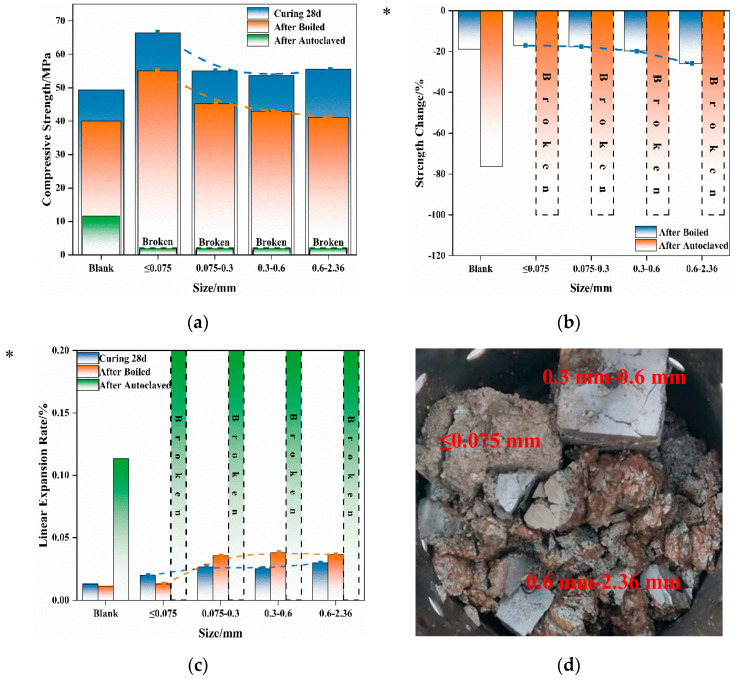
Effect of Mg-Cr brick granules on mortar’s volume stability. (**a**) Mortar strength. (**b**) Strength change of mortar. * Strength change values of broken mortar equal to 100%. (**c**) Linear expansion rate of mortar. * Linear expansion rate values of broken mortar were unavailable. (**d**) Appearance of mortar after the autoclaving treatment.

**Figure 8 materials-16-04675-f008:**
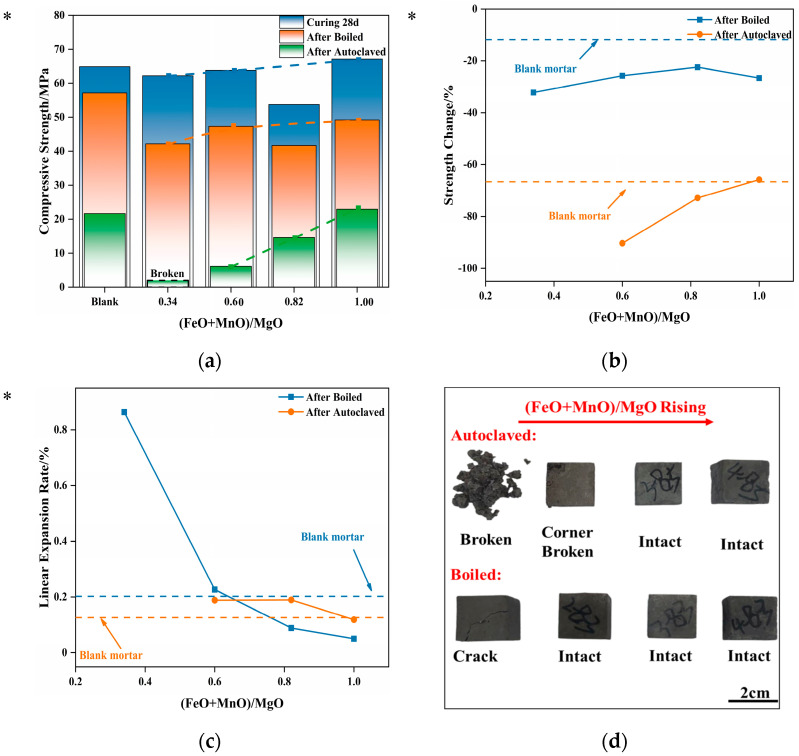
Effect of RO phase granules on mortar’s volume stability. (**a**) Mortar strength. * Mortar was broken when the (FeO + MnO)/MgO = 0.34 after the autoclaving treatment. (**b**) Strength change. * Mortar was broken when the (FeO + MnO)/MgO = 0.34 after the autoclaving treatments; thus, the strength change equals 100%. (**c**) Linear expansion rate. * Mortar was broken when (FeO + MnO)/MgO = 0.34 after the autoclaved treating; thus, the linear expansion rate was unavailable. (**d**) Appearance of tested mortar.

**Figure 9 materials-16-04675-f009:**
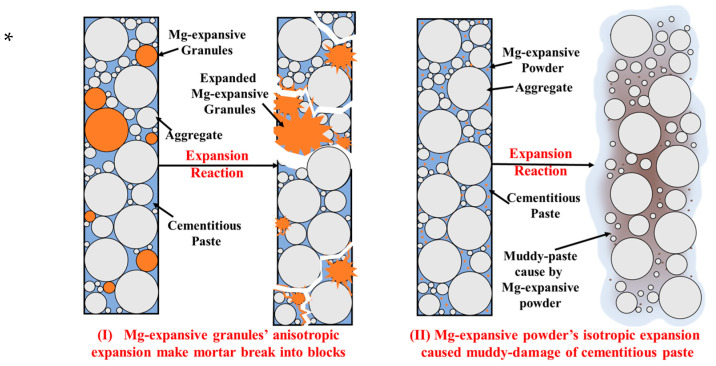
Failure modes caused by Mg-expansive components of different sizes. * For this study, the graph presents the failure modes of the specimens containing 6% waste Mg-Cr refractory brick granule replacement after autoclave treatment.

**Table 1 materials-16-04675-t001:** Standards and corresponding testing methods of steel slag’s volume stability in cementitious materials in China.

Standards	Testing Method	Requirements
GB/T 20491-2017 [16]	Mortar bar + autoclaved test (only tested when the MgO content > 5%)	Linear expansion rate ≤ 0.5%
GB/T 32546-2016 [25]	Mortar bar + autoclaved test	Linear expansion rate ≤ 0.8%
YB/T 4187-2009 [26]	Water immersion test	Immersion expansion rate ≤ 2%
YB/T 4329-2012 [27]	Autoclaved test	Pulverization rate ≤ 5.9%
YB/T 801-2008 [28]	Autoclaved test + water immersion test	Pulverization rate ≤ 5% + immersion expansion rate ≤ 2%
GB/T 25824-2010 [29]	Water immersion test	Immersion expansion rate ≤ 2%
YB/T 4201-2009 [30]	Autoclaved test	Pulverization rate ≤ 5.9%
GB/T 51003-2014 [31]	Mortar bar + autoclaved test (only tested when the MgO content > 13%)	Linear expansion rate ≤ 0.5%
JIS A5015-2018 [32]	80 °C water immersion test	Immersion expansion rate ≤ 2%
ASTM D5106-2013 [33]	70 °C water immersion test	Immersion expansion rate ≤ 0.5%
ASTM D4792-2013 [34]	70 °C water immersion test	Immersion expansion rate ≤ 0.5%
CNS15311-2010 [35]	70 °C water immersion test	Immersion expansion rate ≤ 0.5%

**Table 2 materials-16-04675-t002:** Physical performance of P·O 42.5 cement.

Density (kg/m^3^)	Water Consumption (%)	Specific Surface Area (kg/m^2^)	Setting Time (min)		Flexural Strength (MPa)		Compressive Strength (MPa)	
			Initial	Final	3 d	28 d	3 d	28 d
3010	27	359.2	210	298	5.1	8.2	24.8	47.1

**Table 3 materials-16-04675-t003:** Chemical composition of P·O 42.5 cement and waste Mg-Cr refractory brick (wt%).

Compound	P·O 42.5	Waste Mg-Cr Refractory Brick
SiO_2_	21.98	4.77
Al_2_O_3_	6.02	5.92
CaO	60.05	1.29
Fe_2_O_3_	3.40	8.12
MgO	2.13	58.15
K_2_O	0.64	0.04
Na_2_O	0.15	0.19
ZnO	0.03	0.02
TiO_2_	0.46	0.09
SO_3_	2.09	0.16
P_2_O_5_	0.12	0.10
MnO	0.21	0.28
ZrO_2_	0.02	0.02
Cr_2_O_3_	/	20.51
Loss	2.35	0.08
Other	0.34	0.24

**Table 4 materials-16-04675-t004:** Proportion of RO phase in synthesis.

	Molar Ratio			$\frac{\mathrm{FeO}+\mathrm{MnO}}{\mathrm{MgO}}$	Molecular Mass /g·mol^−1^	Weighing Value of per 100 g End-Product/g			
	MgO	FeO	MnO			MgO/g	Fe/g	Fe_2_O_3_/g	MnCO_3_/g
MgO·0.17FeO·0.17MnO	1.00	0.17	0.17	0.34	64.10	90.98	4.84	13.84	29.89
MgO·0.40FeO·0.20MnO		0.40	0.20	0.60	83.23	70.07	8.95	25.58	27.62
MgO·0.60FeO·0.22MnO		0.60	0.22	0.82	99.02	58.90	11.28	32.25	25.54
MgO·0.75FeO·0.25MnO		0.75	0.25	1.00	111.92	52.11	12.47	35.67	25.68

**Table 5 materials-16-04675-t005:** Atomic percentage of elements in EDS result (%).

Positions	Mg	Fe	Fe/Mg
1	9.53	44.14	4.63
2	10.91	35.96	3.30
3	11.19	20.30	1.81
4	15.66	16.72	1.07
5	32.29	12.81	0.40
6	51.79	/	/

**Table 6 materials-16-04675-t006:** Linear expansion rate and strength change of mortar with the addition of waste Mg-Cr brick granules after the autoclaving treatment.

Replacement	Linear Expansion Rate/%			Strength Change/%		
	≤0.075	0.3–0.6	0.6–2.36	≤0.075	0.3–0.6	0.6–2.36
Blank	0.114			−76.40		
1	0.156	Broken	Broken	−77.16	Broken	Broken
2	Broken	Broken		Broken		
4		Broken				
6		Broken				

## Data Availability

Not applicable.

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
