# Peer review of "Effect of Magnesian-Expansive Components in Steel Slag on the Volume Stability of Cement-Based Materials"

_materials, 2023, doi:10.3390/ma16134675_

Round 1
Reviewer 1 Report
“Effect of Magnesian Expansive Components in Steel Slag on the Volume Stability of Cement-based Materials” will further brighten the effectiveness of steel slag (SS) in the concrete as the authors discussed the advantage and disadvantage of MgO in the SS.
However, the authors need to clarify a few things for the betterment of this work.
Please clarify/define f-MgO and RO phase in line 15; not everyone understood these terms
Define f-CaO (line 56). Though it’s mentioned in the manuscript, I think it will be difficult for those who are not in the field to quickly add them together
Define/write in detail Mg-Cr (line 19)
Line 36-38 is not clear; please reword this sentence.
The lettering in Fig. 8 could be more significant.
Reviewer 2 Report
Line 10, why is the volume stability of steel slag usually ignored?
Lines 14 to 32 of the paragraph can be moved to the introduction part. because we need a clear methodology and also the concluding remarks. so avoid generalized statements.
Line 103, based on a summary of the literature, relates to the present research work as well. The research gap is missing.
line 111. What is P.O. 42.5?
Line 119: "Include fineness of cement P.O. 42.5
Line 122: For Fe2O3 and MnCO3, use subscript wherever required.
Line 122: The presence of iron content leads to corrosion. The author should include durability studies of all proportions to validate the present research work.
Reviewer 3 Report
Reviewer comment: This paper investigates granules made of waste magnesium chromate-based refractory brick (Mg-Cr brick) that come in 14 various sizes and artificial MgOxFeyMnO solid ternary solutions. To examine their impact on the volume stability of cement-based materials, granules with various (FeO+MnO)/MgO ratios were used to replicate magnesian-expansive granules by partially replacing produced sand. Unfortunately, some major revisions were needed:
Comments and suggestions:
1- Please, all abbreviations are required to be checked and revised; we prefer to add the abbreviation list to the manuscript.
2- Please refer to the introduction part, other fundamental questions need to be included in the introduction. So, the author should improve the literature review with recent studies, especially from 2020 until now in Table 1.
3- Please for all equation was mentioned in the manuscript, please cite the cite its used references.
4- Please, fig. 3b is not obvious and all compounds not found. Please double check.
5- Please, on page 11, line 326, please revise and cite the previous studies.
6- It is preferable to add a separate section “recommendations” for future researchers and add some of the important findings to the conclusions and their application.
Reviewer 4 Report
This paper presented a comprehensive investigation about volume stability of cement-based materials due to Magnesian-expansive components. Two sources of Magnesian-expansive components were adopted in the study which were Mg-Cr brick and a synthetic component with different molar ratios.
The paper was organized and written properly. Therefore, it is worth publishing in Materials.
Author Response
Thank you for your affirmation of our work.
Reviewer 5 Report
The article is devoted to the study of the influence of Magnesian-Expansive Components on the volumetric expansion of cement-based materials. There are the following minor remarks:
1) The authors did not indicate what chemical reaction occurs between MgO and cement components (water). By what amount (%) is the resulting reaction product increased in volume?
2) The abstract completely repeats the conclusion. The abstract needs to be rewritten, highlight the most important and make it attractive to researchers.
Reviewer 6 Report
The article is written correctly and is dedicated to interesting research. Nevertheless, some corrections are necessary before its publication. The most important of these is to make linguistic corrections. Some parts of the paper are difficult to understand because of unusual sentence formation or because of grammatical errors. I recommend that the article should be checked by an English native speaker before resubmission.
In addition to this general comment, I also have some specific comments.
1. Figure 1 should be slightly larger to increase its readability. There is still room to make it larger.
2. After printing the article (and one has to assume that someone will want to print it before reading it), it appears that some parts of figure 2 are completely unreadable due to the low contrast and brightness. Particularly those parts that are in the second and third rows in the middle and on the right. It would be good to brighten them up a bit and increase their contrast.
3. In figure 3b, please double the font size and the size of the symbols.
4. In line 276, the authors wrote that samples with four different grain size ranges were tested. Meanwhile, all results are given for only three ranges. The range 0.075-0.3 mm is missing. Please reconcile the stated assumptions with the results.
5. In lines 304-305 the authors write "The regularities (...) are chaotic". In my opinion this is a contradiction. Either we have regularities or chaos.
6. The first conclusion does not follow from the research, because the authors have already written it in the introduction (lines 61-63) citing the work of other researchers. This passage can stay if the authors want to repeat the claims made in it, but it should not be a numbered conclusion, but at most a summary passage preceding the actual conclusions.
In summary, the article is eligible for publication, but it is necessary to make the indicated corrections.
Round 2
Reviewer 2 Report
Is there a standard for boiling treatment? Line 13.
What is your molar ratio, line 16?
Include a waste Mg-Cr refractory brick image in line 154.
Line 166: Due to several implementation-related practical challenges, this idea blends Please factor that into the conventional mix in terms of expense.
general:
1. This study used a few specifics, such as materials, pictures, and the procedure for creating those mortars. Every every step should be documented with a photograph.
2. All findings and conclusions ought to be supported by citations. Many areas have experimental results that have been drafted, but they have not been compared to earlier studies.
Reviewer 3 Report
The manuscript is revised correctly and is acceptable.